# Accumulation of Phenolic Compounds in Microshoot Cultures of *Rhododendron tomentosum* Harmaja (*Ledum palustre* L.)

**DOI:** 10.3390/ijms26167999

**Published:** 2025-08-19

**Authors:** Adam Kokotkiewicz, Sylwia Godlewska, Barbara Sparzak-Stefanowska, Oliwer Panow, Agata Król, Agnieszka Szopa, Mirosława Krauze-Baranowska, Maria Łuczkiewicz

**Affiliations:** 1Department of Pharmacognosy, Faculty of Pharmacy, Medical University of Gdańsk, Hallera 107, 80-416 Gdańsk, Poland; sylwia.godlewska@gumed.edu.pl (S.G.); barbara.sparzak-stefanowska@gumed.edu.pl (B.S.-S.); agata.krol@gumed.edu.pl (A.K.); miroslawa.krauze-baranowska@gumed.edu.pl (M.K.-B.); mlucz@gumed.edu.pl (M.Ł.); 2Department of Medicinal Plant and Mushroom Biotechnology, Faculty of Pharmacy, Jagiellonian University, Medyczna 9, 30-688 Kraków, Poland; a.szopa@uj.edu.pl

**Keywords:** bioflavonoids, 6-(γ,γ-dimethylallylamino)purine, Ericaceae, flavonoid glycosides, flavonols, in vitro shoot culture, Labrador tea, Plantform bioreactor, secondary metabolites, Schenk-Hildebrandt medium

## Abstract

*Rhododendron tomentosum* Harmaja is a marsh plant known for its high content of bioactive components, including essential oil, flavonoids, and phenolic acids. In the current work, the effects of cultivation mode (agar, liquid stationary, shake flask, and temporary immersion) and experiment duration (30, 60, and 90 days) on the growth and contents of non-volatile phenolics in *Rhododendron tomentosum* microshoots were investigated. Agar and liquid stationary cultures provided the highest dry biomass yield per liter, but their dry weight productivities per day were the lowest among the tested systems. Agitated and temporary immersion cultures, on the other hand, were the most productive in terms of fresh and dry biomass yield per day. LC-DAD-ESI-MS analysis of extracts from microshoots and wild-grown plants revealed similarities in phenolic composition: in both cases, the presence of catechin, chlorogenic acid, and flavonoids of flavonol type (mainly glycosides of quercetin and myricetin) was confirmed. The qualitative composition of the phenolic fraction was not affected by experiment duration and cultivation mode. As determined by HPLC analysis, shake flask and temporary immersion cultures were characterized by the highest phenolic contents: up to 37.5 and 26 mg/g dry weight, respectively. The maximum productivities of the above systems were equal to 18 and 13.5 mg/L/d, respectively.

## 1. Introduction

*Rhododendron tomentosum* Harmaja (aka *Ledum palustre* L.) is a small, evergreen shrub, native to the northern regions of Asia and North America, as well as Central and Northern Europe. It is commonly found in arctic and subarctic regions, growing in peaty soils of heaths, marshlands, and pine forests. The plant has been traditionally used for the treatment of several ailments, including rheumatism, asthma, infections of the respiratory tract, gastrointestinal disorders, and skin complaints. Modern studies on *R. tomentosum* demonstrated that extracts of the plant exhibit anti-inflammatory, analgesic, antimicrobial, antifungal, antiviral, anticancer, antidiabetic, and antioxidant properties. The bioactivities mentioned above can be attributed to the presence of a variety of phytochemicals, primarily non-volatile phenolics and essential oil [1].

Given the high therapeutic potential of *R. tomentosum*, in vitro cultures of the species have been previously established in our laboratory to create an alternative platform for the production of bioactive constituents [1]. The studies were focused on developing microshoot cultures, as this type of biomass is usually capable of producing a whole set of phytochemicals typical for the parent species, including volatiles and specialized phenolics [2,3,4,5]. Notably, the established microshoots were shown to be capable of accumulating substantial amounts of essential oil of which the composition differed from the volatile fraction present in an intact plant [1]. The developed cultures also proved to be scalable, as demonstrated by studies involving different bioreactor types, and maintained the ability to biosynthesize essential oil when grown in temporary immersion, airlift, and gas-phase systems [1]. The volatile fraction isolated from the microshoot cultures of *R. tomentosum* showed promising biological activity, exhibiting pro-apoptotic effects on synoviocytes isolated from rheumatoid arthritis (RA) patients [6].

Contrary to essential oil production, the capability of *R. tomentosum* microshoots to biosynthesize non-volatile phenolics remains unknown. Likewise, studies concerning quantitative phenolic composition of the wild-grown *R. tomentosum* are scarce. However, the presence of phenolic compounds, particularly flavonoids, has been linked to anti-inflammatory, antifungal, anticancer, antidiabetic, and antioxidant properties of the plant [1,7,8,9]. Thus, this topic is worthy of further investigation.

Given the above, it was decided to evaluate the potential of *R. tomentosum* microshoots to accumulate phenolic compounds, including phenolic acids, catechin derivatives, and flavonoids. The cultures were maintained in various systems (agar, liquid stationary, shake flask, and temporary immersion bioreactor) for different time periods. In particular, the current study involved long-term cultures of *R. tomentosum* microshoots, maintained for 30, 60, and 90 days. The rationale for the experiments was that, as in vitro shoots are grown for extended time periods, their morphogenic status and capability to accumulate phenolics may change. After evaluating growth parameters, qualitative and quantitative phenolic profiles of the collected biomasses were established using HPLC. For reference, the results were compared with the phenolic composition of aerial parts of the wild-grown *R. tomentosum*, collected from locations in Poland and Finland [6].

## 2. Results and Discussion

### 2.1. Qualitative and Quantitative Analysis of Phenolic Compounds in Wild-Grown Plants and In Vitro Cultures of Rhododendron tomentosum

The MeOH extracts of *R. tomentosum*, prepared as specified in Section 3.4, were analyzed using the method based on the previously validated HPLC-DAD-ESI-MS protocol [10], developed for the separation and quantitation of flavonol glycosides and phenolic acids. The method involved the gradient elution and separation of analytes using a reversed-phase C18 column, as described in detail in the experimental section. An exemplary chromatogram of the phenolic fraction and spectral data of the analytes are presented in Figure 1 and Table 1, respectively. The analysis revealed that the aerial parts of wild-grown plants, originating from Poland and Finland, contain a rich set of phenolics consisting of phenolic acids, flavan-3-ols, and flavonols. Apart from minor differences (Table 1), the qualitative composition of the phenolic fraction in the investigated samples was similar. The identified metabolites involved, among others, gallic (1) and chlorogenic (5) acids, catechin (4), as well as a variety of flavonoid glycosides including the derivatives of quercetin (12–15, 18, 19), myricetin (9, 11, 16, 17), and taxifolin (7). Among the analytes, an interesting constituent was compound 19, of which the DAD and ESI-MS spectra were consistent with esterified quercetin glycoside and showed similarities with the spectral data of tiliroside [11]. The results of LC-DAD-ESI-MS analyses are consistent with previous phytochemical studies on *R. tomentosum* and *R. tomentosum* ssp. *subarcticum,* and the related species *Rhododendron groenlandicum* and *Rhododendron laponicum* [12,13,14,15,16]. The results of the current work, as well as the available literature data, point to the chemotaxonomic importance of the examined compounds. This assertion is supported by the fact that the specimens of *R. tomentosum*, originating from different countries (Poland, Finland, Russia, Canada), were shown to share substantial similarities in terms of phenolic composition. However, it has to be emphasized that the presence of flavonoids, including glycosides of quercetin and myricetin, is a general characteristic of the genus *Rhododendron* [9,17,18]. For instance, compounds of this type have been reported in *Rhododendron przewalskii* [19], as well as numerous other species of *Rhododendrons* found in China [20,21] and the Russian Far East [22]. Given the above, further studies are needed to confirm the chemotaxonomic significance of specific flavonoid constituents in *R. tomentosum*. In particular, experiments should include a larger pool of samples, collected from different geographical locations, and more detailed fingerprint analysis of the compounds of interest. It is also important to acknowledge the limitations of the LC-MS technique employed: several of the analytes were only tentatively identified due to the inability to differentiate between isomeric glycosides. Further studies on *R. tomentosum* chemistry should involve more sophisticated spectroscopic methods, providing insight into the chemical structure of the analytes.

HPLC analysis of MeOH extracts from the basic microshoot culture of *R. tomentosum* demonstrated that the phenolic composition of the in vitro biomass is similar to wild-grown plants (Table 1). Importantly, the presence of flavonoids, primarily from the flavonol group, was validated. The microshoots retained the ability to accumulate metabolites, which may contribute [1,7,8,9] to their biological properties. Given that the investigated biomass was previously shown to produce essential oil [1] with documented biological activity [1,6], the synergism of action of phenolics and volatile compounds cannot be excluded, and may be investigated in future bioactivity studies. Based on the results of phytochemical analysis, *R. tomentosum* microshoots were assigned to further biotechnological experiments, focused on selecting a growth system and culture duration optimal for the accumulation of non-volatile phenolics, particularly of the flavonoid type.

### 2.2. Biomass Growth and Phenolics Accumulation in Rhododendron tomentosum Microshoot Cultures Maintained in Different In Vitro Systems

The effects of cultivation mode and experiment duration on growth parameters of *R. tomentosum* microshoots were presented in Figure 2. Agar cultures provided the highest FW yield over the course of 60–90 days, equal to ca. 320–330 g/L. These values corresponded to high Gi_FW_ rates of ca. 900% and FW productivity of 3.75–5.3 g/L/d. As far as DW content is concerned, the agar culture yielded up to ca. 17.5–20 g/L, corresponding to a Gi_DW_ of ca. 900% and DW productivity of 0.2–0.32 g/L/d. For comparison, the liquid stationary culture yielded a somewhat lower FW (up to 270 g/L), GiFW (up to 760%), and FW productivity (up to ca. 4 g/L/d). However, the DW parameters of the discussed culture were comparable to agar-grown microshoots, which is indicative of a higher water content of the latter. Both stationary cultures remained viable and did not show signs of necrosis over the entire experiment (Figure 3 and Figure A1). Compared to the microshoots grown in stationary media, the agitated culture was characterized by lower FW (content up to ca. 230 g/L and Gi_FW_ up to 650%) and DW parameters (up to 17 g/L, corresponding to Gi_DW_ of 750%) within a 60–90 day period. On day 30, however, the agitated culture yielded the highest amounts of FW (ca. 200 g/L) and DW (ca. 17 g/L), corresponding to Gi_FW_ and Gi_DW_ indices of 575 and 800%, respectively. The FW and DW productivities on the 30th day were the highest among all the experiments, and equaled 6.75 and 0.56 g/L/d, respectively. On the other hand, as the experiment progressed, the recorded productivities of the agitated culture dropped significantly, reaching 2.5 g/L/d FW and 0.16 g/L/d DW on day 90, the lowest values over the course of the whole experiment.

The results confirm that cultivation time and the type of growing system are essential factors affecting biomass yield and secondary metabolite content of plant in vitro cultures. Similar experiments have been previously conducted using the microshoot cultures of several medicinally relevant species, such as *Scutellaria baicalensis* [23], *Schisandra chinensis* [24], *Salvia apiana* [25], and *Leucojum aestivum* [26]. The current work was specifically designed to investigate the effects of long-term cultivation (30–0–90 days) on the growth and phenolic composition of *R. tomentosum* microshoots, maintained in systems varying with respect to medium type (agar or liquid one), mixing (agitation or lack thereof), and aeration mode. All experiments were conducted using a 2iP-supplemented SH medium, which provided stable growth of the examined biomass. The microshoots grown in said medium branched out intensively but also elongated at the same time, uniformly filling the headspace of growth containers in the course of the experiments (Figure 3). On the other hand, the thidiazuron-supplemented media used previously provided intensive branching, but unfortunately resulted in stunted shoots [27], a phenomenon frequently observed during micropropagation experiments [28], including studies on *Rhododendron* plants [29]. The growth vessels used in the current work were selected based on the headspace size: both Magenta vessels and Plantform bioreactor [30] provide sufficiently large overhead volume, which was meant to accommodate elongated *R. tomentosum* shoots over extended time periods. For the same reason, the more popular [31,32] RITA bioreactor was not included in the experiments, although it was successfully used in previous studies on *R. tomentosum* [33]. Compared to the Plantform system, the available volume in RITA bioreactors was noticeably smaller [31,34].

The results indicate that while the stationary cultures of *R. tomentosum* provide high biomass yields, their productivity in this regard is low due to more extended time periods required to grow the microshoots. The agitated culture, on the other hand, provided faster growth and high productivity within a 4-week period, but declined noticeably faster. As shown in Figure 3 and Figure A1, microshoots grown in shake flasks for 90 days showed visible signs of necrosis in a submerged part of the biomass: a clear indicator of the culture entering a decline phase. A moderate medium and explant browning were also observed in the 90-day-old liquid stationary culture (Figure 3), but it was not correlated with a decreased biomass yield. Importantly, good results in terms of biomass growth were obtained using a temporary immersion bioreactor. Similar to the agitated cultures, the system provided high biomass yield within a 30-day period (DW content equal to ca. 14 g/L, Gi_DW_ of ca. 650% and DW productivity of 0.475 g/L/d). As depicted in Figure 2, the recorded growth parameters were higher than in the stationary cultures, yet lower compared to the agitated system. Similar to the other liquid medium systems investigated, medium browning was noticeable on the 90th day of the experiment, but the microshoots themselves did not show signs of necrosis (Figure 3).

Overall, the conducted experiments demonstrated that the microshoot cultures of *R. tomentosum* are characterized by relatively high growth rates. The recorded growth indices (up to ca. 900%, Figure 2) were comparable to the microshoot cultures of other plants, such as *Securinega suffruticosa* [35], *Schisandra chinensis* [36], and *Salvia apiana* [25], previously investigated by the authors. Unfortunately, valid comparisons of microshoots’ growth rates with reports by other research teams are often not possible. In many cases, biomass yields are either not provided or expressed as the number/weight of individual shoots/explants. As demonstrated by the current work, medium mixing and aeration clearly contributed to higher productivity of the tested systems, as shown in agitated and temporary immersion cultures. This observation is in agreement with the results of comparative studies involving microshoot cultures of other species. For instance, the agitated microshoots of *S. chinensis* were characterized with higher growth rates than agar and liquid stationary cultures of the plant [36]. In the other work, microshoots of *S. chinensis* maintained in temporary immersion systems for 60 days provided biomass yield comparable to the agitated culture; however, the latter showed higher biomass productivity over the shorter 30-day period [24]. Similar trends were observed in *S. apiana*, of which the microshoots grown as an agitated culture reached the growth maximum earlier (and subsequently declined), whereas a culture maintained in a temporary immersion system was characterized by noticeably longer stationary phase [25]. Likewise, the in vitro shoots of *Scutellaria baicalensis* showed higher growth rates when grown in a temporary immersion system, as compared to stationary cultures [23]. Another example is the in vitro shoots of *Leucojum aestivum* of which the growth rates in a custom-made temporary immersion system exceeded the values recorded for the stationary culture [26].

Besides being examined for growth rates, the microshoot cultures of *R. tomentosum* were investigated for the content of non-volatile phenolics. The qualitative HPLC analysis demonstrated that regardless of the cultivation mode and experiment duration, the phenolic profile of the investigated culture remained virtually unchanged, and included several flavonoids (chiefly of flavonol type) accompanied by catechin and phenolic acids. The average contents of major metabolites in the in vitro biomasses and the reference intact plant material are included in Table 2. On the other hand, Figure 4 illustrates the influence of culture type and harvest time on total phenolic content (expressed as mg/g DW and mg/L) and productivity (expressed as mg/L/d) of the microshoot cultures. Additionally, contents of the respective phenolic compounds in *R. tomentosum* microshoots under different experimental conditions were presented in Figure 5. The experiments demonstrated that the agitated culture was characterized by the highest phenolic content per gram DW (32.5 and 37.5 mg on days 30 and 60, respectively) and liter of culture (560 and 625 mg on days 30 and 60, respectively). The productivity of the shake flask culture was also the highest, reaching ca. 19 mg/L/d on day 30. High phenolic contents (29 mg/g DW) and productivities (14 mg/L/d) were also recorded for the 30-day microshoots maintained in a temporary immersion system. As depicted in Figure 4, phenolic accumulation and production parameters were noticeably lower for the agar and stationary liquid cultures of *R. tomentosum*. Another important observation is that regardless of culture type, the 90-day old microshoots had a noticeably lower phenolic content than those harvested on day 30 or 60, which resulted in low productivities of the systems (Figure 4). This was particularly visible in the case of the shake flask culture of which the productivity dropped to ca. 1 mg/L/d after 90 days of cultivation. The likely cause of that was biomass necrosis; however, the assessment of enzyme activities (such as phenolic synthase) would be necessary to provide a more comprehensive explanation of the phenomenon. It is also worth noting that for most of the experimental variants, the sum of phenolics (determined by HPLC) in the microshoot culture was higher compared to the intact plant material (Table 2, Figure 4). This was chiefly due to the presence of substantial amounts of catechin (4) (Table 2, Figure 5) of which the concentration in agitated cultures was up to ca. 32 mg/g DW. In the wild-grown plants, the compound was found at a low level, below the quantitation limit. Other compounds whose contents in the microshoots exceeded those in intact plant material were quercetin 3-rhamnoside (15), myricetin pentoside (isomer 2) (16), myricetin (17), and quercetin glycoside (19). Protocatechuic acid (2), chlorogenic acid (5), taxifolin hexoside (7), quercetin 3-O-galactoside (10), quercetin 3-O-xylopyranoside (12), quercetin 3-O-arabinoside (13), quercetin pentoside (14), and quercetin acetylglycoside/acetylgalactoside (18), on the other hand, prevailed in the wild-grown plants (Table 2, Figure 5). Overall, concentrations of the above listed compounds did not exceed 2.5 mg/g DW. The above comparisons give an idea regarding the performance of the established cultures with respect to the production of phenolic compounds. However, they should be treated with caution as they do not take into account inter- and intraseasonal variations in the contents of the specific metabolites in wild-grown plants. For instance, concentrations of the respective quercetin glycosides in *R. groenlandicum* were shown to vary by 20–120% between years, whereas the inter-seasonal difference in total phenolics was ca. 30% [15]. Intraseasonal variations were also observed, as demonstrated in studies on *R. tomentosum* ssp. *subarcticum* [12]: between June and September, the concentrations of individual phenolics differed by up to ca. 100%. Given the above, the established in vitro cultures are not necessarily a richer source of phenolics of interest. However, they can be regarded as more stable in this regards, as they are maintained in a controlled environment and are not expected to exhibit seasonal fluctuations in metabolite content.

The results of the current work were compared with previous studies dealing with the production of secondary metabolites in microshoot cultures, maintained in different types of in vitro systems. The available data indicate that the accumulation of bioactive compounds is generally higher in cultures provided with better aeration. For instance, the in vitro shoots of *S. baicalensis* accumulated significantly higher amounts of bioactive flavonoids (including baicalin and baicalein) when grown in a temporary-immersion system, compared to an agar culture [23]. Similarly, the in vitro shoots of *Pueraria tuberosa* were shown to produce higher amounts of isoflavonoids when grown as an agitated culture, in comparison to a liquid stationary one [37]. The authors also demonstrated that biosynthesis of isoflavones can be increased by providing aeration to *P. tuberosa* shoots, maintained as a liquid stationary culture in the Growtek bioreactor [37]. As far as *S. chinensis* microshoots are concerned, the cultivation mode (agar, liquid stationary, or agitated) did not affect their capability to accumulate dibenzocyclooctadiene lignans [36]. However, levels of bioactive lignans were noticeably higher in *S. chinensis* microshoots grown in temporary immersion systems, which points to aeration as a crucial factor contributing to increased production of said metabolites [24]. Studies on *L. aestivum* indicate that parameters such as medium volume, as well as aeration and immersion frequencies, determine the effectiveness of a temporary immersion system [26]. The experiments demonstrated that agitation alone did not contribute to increased galanthamine production, compared to the liquid stationary culture. Depending on the configuration and settings of the temporary immersion systems employed, the production of galanthamine was lower, higher, or comparable to liquid stationary and agitated culture [26].

Notably, the conducted experiments demonstrated that *R. tomentosum* microshoots retained the capability to accumulate a noticeable amount of phenolics when grown in a temporary-immersion bioreactor. In terms of productivity, the TIS system was second to the agitated culture (Figure 4), but compared to the latter, it offers several advantages [2]. Agitated cultures can be considered as a gold standard in basic experiments involving medium optimization, plotting growth curves, determining inoculum size, etc. However, they cannot be scaled up and, besides setting the agitation speed, offer limited possibilities of modifying the aeration rate. An important disadvantage of shake flask cultures is permanent submersion of plant explants, which often results in shoot hyperhydricity and poor growth, especially in cytokinin-supplemented media [38,39]. Because of their morphology and light dependence, in vitro shoot cultures are inherently challenging to be scaled-up to serve as a source of specialized metabolites. In this regard, TIS installations are relatively simple and cheap, and at the same time offer more control of a bioprocess over shake flask cultures. Importantly, they enable optimization of the aeration rate and submersion frequency, facilitating proper shoot morphology, maximum growth, and/or metabolite content. Nowadays, numerous TIS systems are commercially available which enables to select the bioreactor model suitable for the investigated culture [4,32,39,40,41]. The disadvantage of TIS systems is that individual containers are still relatively small. However, to increase biomass production, multiple TIS bioreactors are stacked in growth chambers instead of employing a conventional seed train as in the case of cell suspension cultures [2].

An important observation is that both shake flask and the TIS system used in the study favored shorter cultivation times, as demonstrated by the highest productivities recorded for 30-day cultures (Figure 4). Given the above, further studies should focus on plotting detailed growth curves of both culture types, with the aim of determining exact growth and production maxima. The agar culture, on the other hand, provides more stable growth and secondary metabolite content over extended time periods, and can thus serve as a reservoir of biomass for future biotechnological and phytochemical studies.

## 3. Materials and Methods

### 3.1. Reagents

The reagents used in plant in vitro culture experiments were from Sigma-Aldrich (St. Louis, MO, USA). The methanol (pure p. a. grade) used for extracts’ preparation was from POCH (Gliwice, Poland). The LC-MS-grade acetonitrile used in HPLC analysis was from Merck (Darmstadt, Germany). Type I water was produced using an Ultra UV system (Hydrolab, Straszyn, Poland). The reference substances used in HPLC analysis were obtained from the following: Sigma-Aldrich (gallic acid, protocatechuic acid); Fluka, Buchs, Switzerland (catechin, p-hydroxybenzoic acid, hyperoside, myricetin, quercetin); ChemFaces, Wuhan, China (quercetin 3-O-xylopyranoside); PhytoLab, Vestenbergsgreuth, Germany (quercetin 3-O-arabinoside); and Extrasynthèse, Genay, France (chlorogenic acid).

### 3.2. Plant Material

Microshoot cultures were previously obtained from vegetative aerial parts of *Rhododendron tomentosum*, collected near Miszewko, Poland [1]. They are further referred to as the ‘basic culture’ and served as a source of plant material for in vitro experiments. The cultures were maintained in baby food jars (Sigma-Aldrich) on a modified Schenk–Hildebrandt (SH) medium, supplemented with 9.84 µM 6-(γ,γ-Dimethylallylamino)purine (2iP), 30.0 g/L sucrose, and 6.0 g/L agar. The cultures were grown at 24 ± 2 °C under continuous light (40 μmol m^−2^ s^−1^ white fluorescent light, TLD 35 W/33 tubes, Philips, Amsterdam, The Netherlands) and subcultured at 30-day intervals.

The aerial parts (leaves and stems) of wild-grown *R. tomentosum*, serving as reference intact plant material in phytochemical analysis, were collected in June 2016 near Miszewko (Pomeranian Voivodeship, Poland), from the same specimen that was used for in vitro culture initiation. The study also included aerial parts of wild-grown plant collected in June 2016 near Turku (Southwest Finland), as detailed earlier [6]. The harvested plants were air-dried and kept desiccated in the dark in sealed containers.

### 3.3. Experimental In Vitro Cultures

All in vitro culture experiments were conducted under the conditions specified above, using modified SH medium of the same composition (Section 3.2). The inoculum was obtained from a basic culture of *R. tomentosum* microshoots, grown for 30 days. The experimental cultures were inoculated at a biomass-to-medium ratio of 3/100 (*w*/*v*), and maintained for 30, 60, and 90 days.

Agar and stationary liquid cultures were grown in Magenta^TM^ vessels (77 mm × 77 mm × 97 mm, Sigma-Aldrich, St. Louis, MO, USA), filled with 100 mL of stationary or liquid medium, respectively. In liquid stationary cultures, a support from stainless steel mesh (1 × 1 mm) was placed 13 mm above the bottom. Agitated cultures were maintained in 250 mL Erlenmeyer flasks containing 100 mL of a liquid medium, closed with silicone sponge stoppers (Carl-Roth, Karlsruhe, Germany), placed on a rotary shaker (120 rpm, 25.4 mm orbit, INNOVA 2300, Eppendorf, Hamburg, Germany). Temporary-immersion cultures were grown in the Plantform bioreactor (Plant Form AB, Hjärup, Sweden), filled with 500 mL of the medium, with an immersion cycle set at 5 min every 90 min. The aeration rate was 1 *vvm* (air volume/medium volume/minute).

The harvested cultures were evaluated for fresh weight (FW) content and frozen at −20 °C. The samples were freeze-dried (1 × 10^−1^ mbar, 72 h, Steris Lyovac GT2 lyophilizer; Finn-Aqua Santasalo-Sohlberg, Tuusula, Finland) and subsequently evaluated for dry weight (DW) content.

### 3.4. Extract Preparation

Approximately 1 g samples of pulverized, dried (wild-grown plants) or freeze-dried (in vitro cultures) plant material were extracted (3 × 50 mL MeOH, 3 × 60 min) at room temperature using a laboratory shaker (100 rpm, WL-2000, JW Electronics, Warsaw, Poland). The MeOH extracts were pooled, evaporated in vacuo (190 mbar, water bath set at 50 °C, 18 min distillation time; R-300 evaporator, Büchi, Flawil, Switzerland). The dry residues were dissolved in MeOH, and the volume was made up to 25 mL. The prepared samples were stored at 8 °C. Prior to HPLC analysis, the extracts were centrifuged for 10 min at 14,000 rpm (65R centrifuge, MPW, Warsaw, Poland).

### 3.5. HPLC Analysis

The HPLC analysis was conducted using a Shimadzu LC chromatograph (Kyoto, Japan). The system consisted of the following: two pumps (LC-20AD), a degasser (DGU-20A5), a semi-micro mixer, a system controller (CBM-20A), a column thermostat (CT0-20AC), an autosampler (SIL-20ACXR), a photodiode array detector (SPD-M20A), and a mass spectrometer with ESI ionization (LCMS-2020). The data were recorded and processed using Lab Solution software (Version 5.86 SP1, Shimadzu, Kyoto, Japan). The analysis followed the previously published and validated method [10]. Separations were carried out on a reverse-phase column (Kinetex C18, 100 mm × 2.1 mm, 2.6 μm; Phenomenex, Torrance, CA, USA), using gradient elution. The mobile phase consisted of the following: A—H_2_O:FA 1000:1 (*v/v*) and B—H_2_O:MeCN:FA 500:500:1 (*v*/*v*/*v*). The gradient program was as follows: 0 min, 12% B; 10 min, 20% B; 35 min, 43% B; 60 min, 100% B; 64 min, 100% B; 66 min, 12% B; 81 min, 12% B; end. The flow rate was 0.2 mL/min, column temperature 20 °C, injection volume 1 μL (quantitative analysis) or 2–4 μL (qualitative analysis). LC–DAD data were recorded over the 190–800 nm range. Mass spectrometric detection was performed in both negative (N) and positive (P) ion modes over two mass ranges: *m*/*z* 100–350 and *m*/*z* 200–800, using scan-type acquisition. The parameters of electrospray ionisation were as follows: heating temp. 200° C; desolvation line temp. 250 °C, nebulizing gas flow: 1.5 L/min, drying gas flow 15 L/min., detector voltage: 1.45 kV (N) or 1.3 kV (P). Peaks in chromatograms were identified by comparison of their retention times with those of standards, and/or analysis of the recorded LC-DAD and LC-ESI-MS spectra. Analytes were quantified using external standards, based on the peak areas recorded at 254 nm. Standard calibration curves were plotted using dilution series of catechin (quantitation of flavan-3-ols), chlorogenic acid (quantitation of phenolic acids), and quercetin (quantitation of flavonoids).

## 4. Conclusions

The presented work demonstrated that the microshoot cultures of *R. tomentosum* are capable of accumulating phenolic compounds typical for the parent plant, including flavan-3-ols and flavanols. The experiments proved that both the type of in vitro system used and cultivation time affect secondary metabolite content of the investigated biomass, with temporary immersion culture being the most promising in terms of phenolic productivity and future prospects. The collected data indicate that the established cultures can be considered an alternative source of a variety of flavonol glycosides, including derivatives of quercetin and myricetin. Compared to the respective aglycones, compounds of this group are less abundant, which encourages further studies focused on developing downstream procedures for their isolation and purification. Future projects will also include evaluation of antioxidant and anti-infammatory activity of the extracts and specific metabolites obtained from the developed culture. From a biotechnological standpoint, *R. tomentosum* microshoots are a suitable platform for elicitation experiments [1]; further studies in this area will include determining the effects of stress factors on flavonoid accumulation.

Despite promising results, it is crucial to point out the limitations of the current work. The phytochemical experiments should be extended and include a more comprehensive approach to fully elucidate the structures of the tentatively identified analytes. The biotechnological part of the work requires further optimization of the cultivation time of microshoots maintained in different systems, particularly the temporary immersion bioreactor, to maximize biomass yield and phenolic productivity. Systematic studies on the effects of growth medium components, particularly plant growth regulators, could also provide valuable data. Moreover, a more mechanistic approach is required to explain the observed phenomena, related to growth and secondary metabolite production in the microshoots. With this in mind, assessing the activity of enzymes such as phenolic synthase, as well as those related to oxidative stress, would be desirable.

## Figures and Tables

**Figure 1 ijms-26-07999-f001:**
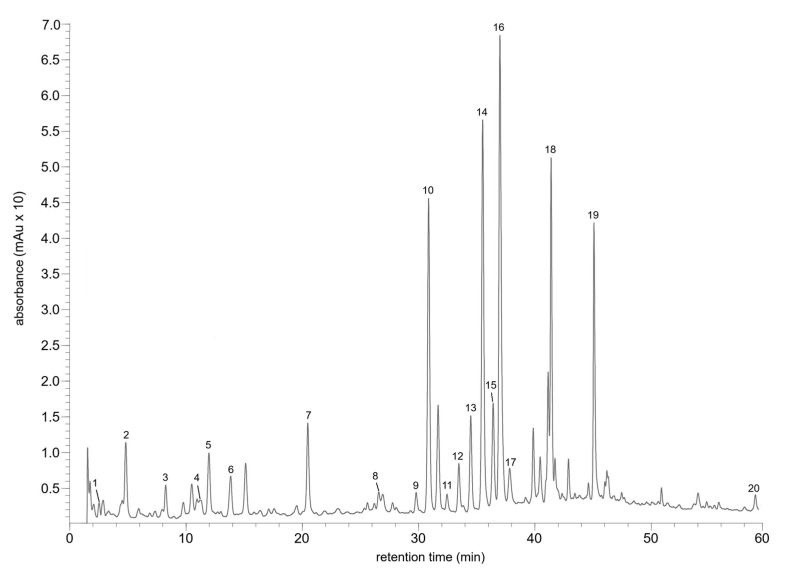
HPLC chromatogram (λ = 254 nm) of the phenolic fraction in MeOH extract from aerial parts of wild-grown *R. tomentosum* (Miszewko). Peak numbers correspond to Table 1.

**Figure 2 ijms-26-07999-f002:**
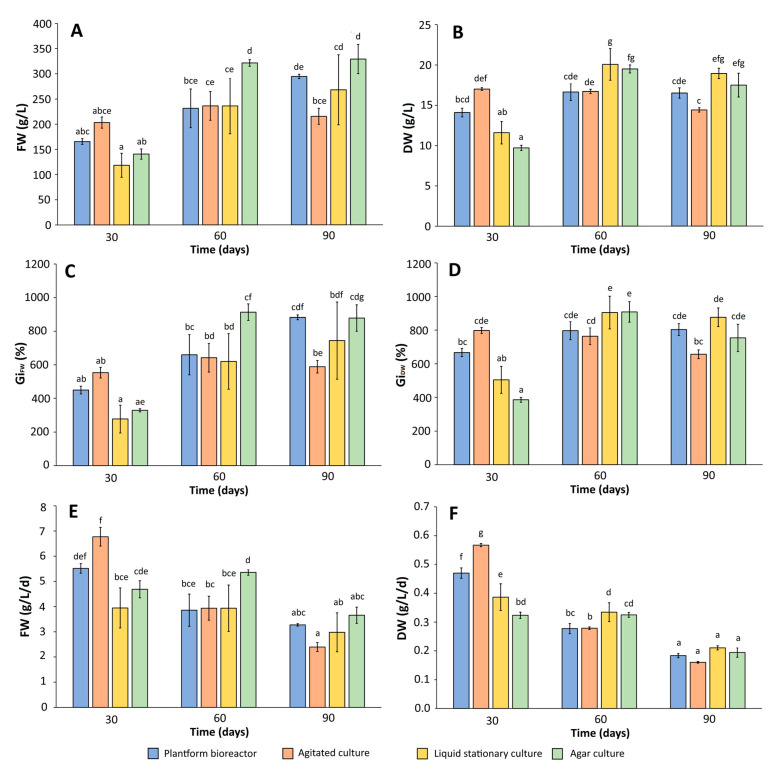
The influence of culture type and experiment duration on the growth parameters of *Rhododendron tomentosum* microshoots: (**A**) fresh weight (g/L); (**B**) dry weight (g/L); (**C**) growth index for fresh weight (%); (**D**) growth index for dry weight (%); (**E**) fresh weight productivity (g/L/d); and (**F**) dry weight productivity (g/L/d). Growth indices were calculated according to the following formulas: Gi_FW_ = [(FW_1_ − FW_0_) ÷ FW_0_] × 100%, where FW_1_ is the fresh weight of the harvested biomass (g) and FW_0_ is the fresh weight of the inoculum (g); Gi_DW_ = [(DW_1_ − DW_0_) ÷ DW_0_] × 100% where DW_1_ is the dry weight of the harvested biomass (g) and DW_0_ is the dry weight of the inoculum (g). The presented values are the means of three samples ± SD. Different letters indicate significant differences between the means at *p* < 0.05 (one-way ANOVA followed by Tukey’s post hoc test).

**Figure 3 ijms-26-07999-f003:**
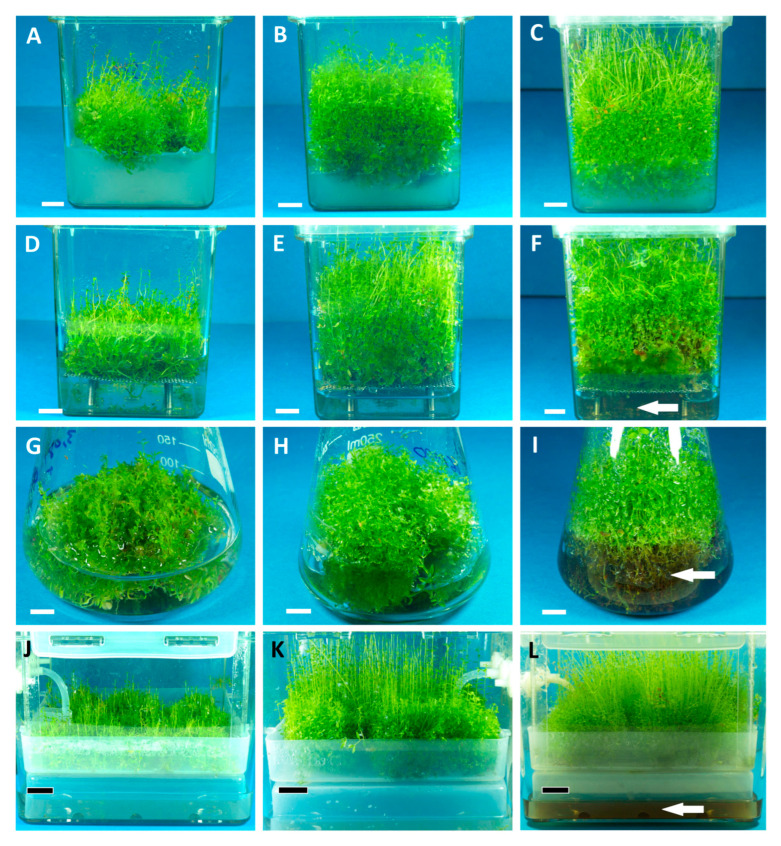
Different types of *Rhododendron tomentosum* microshoot cultures: agar culture grown for 30 (**A**), 60 (**B**), and 90 days (**C**); liquid stationary culture grown for 30 (**D**), 60 (**E**), and 90 days (**F**); agitated/shake flask culture grown for 30 (**G**), 60 (**H**), and 90 days (**I**); temporary-immersion (Plantform bioreactor) culture grown for 30 (**J**), 60 (**K**), and 90 days (**L**). White bar = 1 cm, black bar = 3 cm. White arrows indicate signs of culture necrosis (explant and medium browning).

**Figure 4 ijms-26-07999-f004:**
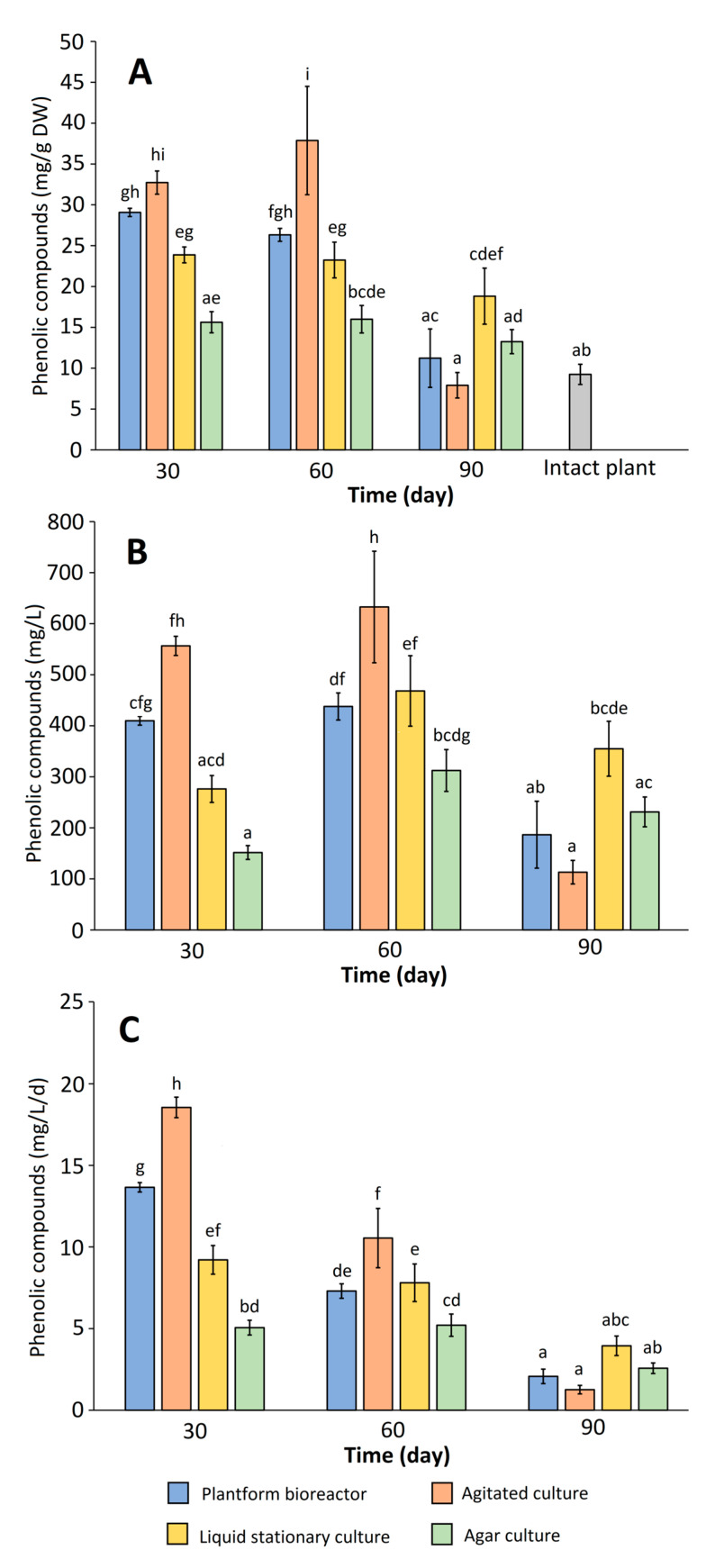
The influence of culture type and experiment duration on total content of the selected phenolics in *Rhododendron tomentosum* microshoots, determined by HPLC: (**A**) phenolic content expressed as mg/g DW; (**B**) phenolic content expressed as mg/L; (**C**) phenolic productivity expressed as mg/L/d. The presented values are the means of three samples ± SD. Different letters indicate significant differences between the means at *p* < 0.05 (one-way ANOVA followed by Tukey’s post hoc test). Intact plant: aerial plants of wild-grown *R. tomentosum*, harvested near Miszewko, Poland.

**Figure 5 ijms-26-07999-f005:**
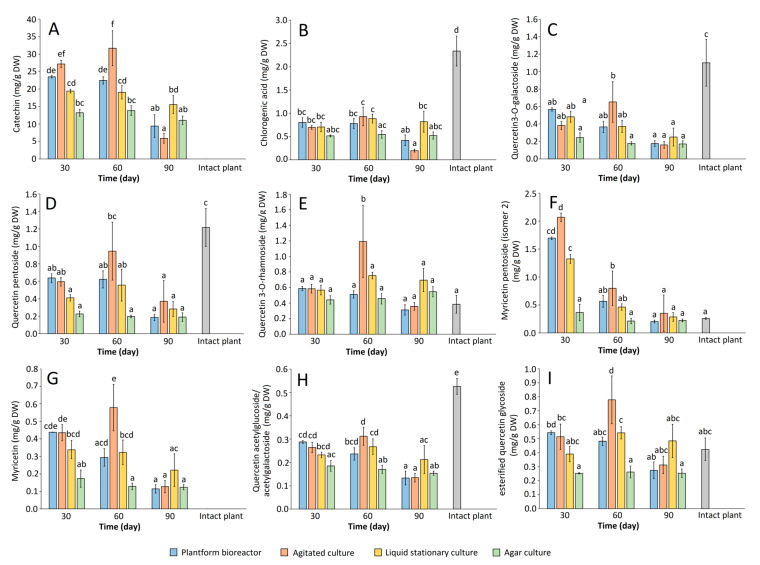
The influence of culture type and experiment duration on contents (expressed as mg/g DW) of the selected phenolics in *Rhododendron tomentosum* microshoots, determined by HPLC: (**A**) catechin; (**B**) chlorogenic acid; (**C**) quercetin 3-O-galactoside; (**D**) quercetin pentoside; (**E**) quercetin 3-O-rhamnoside; (**F**) myricetin pentoside (isomer 2); (**G**) myricetin; (**H**) quercetin acetylglucoside/acetylgalactoside; (**I**) quercetin glycoside. Different letters indicate significant differences between the means at *p* < 0.05 (one-way ANOVA followed by Tukey’s post hoc test). Intact plant: aerial plants of wild-grown *R. tomentosum*, harvested near Miszewko, Poland.

**Table 1 ijms-26-07999-t001:** LC-MS and LC-DAD data of phenolic constituents of wild-grown plants and the microshoot cultures of *Rhododendron tomentosum*. Numbering corresponds to the chromatogram in Figure 1. ND = not detected.

No.	t_R_ (min)	Λ_max_ (nm)	[M + H]^+^	[M − H]^−^	Compound	Wild-Grown Plant		Microshoot Culture
						Miszewko	Finland	
1	2.52	209, 269	-	169	Gallic acid ^a^	+	ND	ND
2	4.82	207, 251, 294	-	153	Protocatechuic acid ^a^	+	+	ND
3	8.26	204, 254	-	-	*p*-Hydroxybenzoic acid ^a^	+	ND	ND
4	11.28	212, 278, 318	291	289	Catechin ^a^	+	+	+
5	11.96	294sh, 325	355	353	Chlorogenic acid ^a^	+	+	+
6	13.02	209, 229, 301sh, 338	209	369	Chlorogenic acid methyl ether ^b^	+	+	+
7	20.46	197, 230sh, 289, 339sh	305, 467	465	Taxifolin hexoside ^b^	+	+	+
8	26.55	201, 226sh, 277	577	575	Procyanidin A1/A2 ^b^	+	+	+
9	29.77	206, 257, 303sh, 351	451, 319	449	Myricetin pentoside (isomer 1) ^b^	+	ND	ND
10	30.85	200, 252, 267sh, 300sh, 353	465, 303	463	Hyperoside ^a^	+	+	+
11	32.45	206, 251, 367	481, 319	479	Myricetin galactoside ^b^	+	ND	ND
12	33.46	204, 254, 266sh, 300sh, 352	435, 303	-	Quercetin 3-O-xylopyranoside ^a^	+	+	ND
13	34.49	202, 255, 265sh, 300sh, 353	435, 303	-	Quercetin 3-O-arabinoside ^a^	+	+	+
14	35.51	202, 254, 264sh, 350	435, 303	-	Quercetin pentoside ^b^	+	+	+
15	36.40	202, 254, 264sh, 300sh, 346	449, 303	447	Quercetin-3-rhamnoside ^b^	+	+	+
16	36.99	207, 252, 268sh, 300sh, 368	451, 319	449	Myricetin pentoside (isomer 2) ^b^	+	+	+
17	37.82	204, 257, 300sh, 369	319	317	Myricetin ^b^	+	ND	+
18	41.39	202, 254, 266sh, 300sh, 350	507, 303	505	Quercetin acetylgalactoside/acetylglucoside ^b^	+	+	+
19	45.08	197, 256, 266sh, 314, 363sh	611, 303	-	Esterified quercetin glycoside ^b^	+	+	+
20	58.95	266, 333	285	283	6.7-dihydroxy-4′-methoxyisoflavone ^b^	+	+	ND

^a^ Compound identified based on co-chromatography with a standard substance, and comparison of LC-DAD and LC-ESI-MS spectra. ^b^ Compound tentatively identified based on the analysis of LC-DAD and LC-ESI-MS spectra, and their comparison with the literature data.

**Table 2 ijms-26-07999-t002:** Contents of the selected phenolic metabolites (mg/g DW) in intact plant material and microshoot cultures of *Rhododendron tomentosum*. Values represent average contents (*n* = 3). Numbering corresponds to the chromatogram in Figure 1.

No.	Compound	Plantform Bioreactor			Agitated Culture			Liquid Stationary Culture			Agar Culture			Intact Plant (Miszewko)
		30 d	60 d	90 d	30 d	60 d	90 d	30 d	60 d	90 d	30 d	60 d	90 d	
2	Protocatechuic acid	-	-	-	-	-	-	-		-	-	-	-	1.97
4	Catechin	23.51	22.47	9.41	27.18	31.68	5.90	19.42	19.08	15.56	13.22	13.84	11.07	-
5	Chlorogenic acid	0.80	0.78	0.42	0.70	0.93	0.20	0.71	0.89	0.82	0.51	0.55	0.52	2.34
7	Taxifolin hexoside	-	-	-	-	-	-	-	-	-	-	-	-	0.43
10	Hyperoside (quercetin 3-O-galactoside)	0.57	0.37	0.18	0.38	0.65	0.16	0.48	0.37	0.25	0.24	0.18	0.17	1.10
12	Quercetin 3-O-xylopyranoside	-	-	-	-	-	-	-	-	-	-	-	-	0.27
13	Quercetin 3-O-arabinoside	-	-	-	-	-	-	-	-	-	-	-	-	0.32
14	Quercetin pentoside	0.64	0.62	0.19	0.60	0.95	0.37	0.41	0.56	0.28	0.23	0.20	0.19	1.22
15	Quercetin-3-rhamnoside	0.59	0.51	0.31	0.58	1.19	0.36	0.57	0.75	0.69	0.44	0.46	0.55	0.38
16	Myricetin pentoside (isomer 2)	1.70	0.56	0.20	2.07	0.80	0.35	1.32	0.46	0.29	0.37	0.21	0.22	0.26
17	Myricetin	0.44	0.29	0.11	0.43	0.58	0.13	0.34	0.32	0.22	0.17	0.13	0.12	-
18	Quercetin acetylgalactoside/ acetylglucoside	0.29	0.24	0.13	0.26	0.31	0.13	0.23	0.27	0.21	0.19	0.17	0.15	0.53
19	Quercetin glycoside	0.54	0.48	0.27	0.51	0.78	0.31	0.39	0.54	0.48	0.25	0.26	0.25	0.42
	Sum	29.07	26.32	11.23	32.72	37.87	7.91	23.87	23.24	18.82	15.63	15.99	13.25	9.24

## Data Availability

The original data presented in the study are openly available in Polish, at the Platform of Medical Research at http://dx.doi.org/10.60816/0k1b-9850 (accessed on 14 July 2025).

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
