# Peer review of "Accumulation of Phenolic Compounds in Microshoot Cultures of Rhododendron tomentosum Harmaja (Ledum palustre L.)"

_ijms, 2025, doi:10.3390/ijms26167999_

Round 1
Reviewer 1 Report
Comments and Suggestions for Authors
This study investigated the effects of different culture modes and experimental durations on the growth of Rhododendron tomentosum microshoots and the synthesized content of non-volatile phenolics. The authors measured the phenolic content and a series of other physiological parameters (e.g. fresh weight, dry weight, etc.) of Rhododendron tomentosum microshoots under agar, liquid, shaking flask and temporary immersion cultivation systems for 30, 60 and 90 days. Comparison with the wild-grown plants, the data revealed that higher productivity of shake flask culture versus temporary immersion culture, and phenolic metabolism analyses showed stable synthesis of the same compounds (catechins, chlorogenic acids, and quercetin/populin glycosides) in all the systems, with shake flask culture having the highest phenolic content, followed by temporary immersion culture. This study revealed the key role of dynamic aeration on phenolic synthesis in Rhododendron tomentosum microshoots, and provided theoretical and practical basis for the optimization of in vitro production system of medicinal plants.
However, there are still some doubts in the study:
- The phenolic content was measures, however, the differences of biological activities (e.g. antioxidant and anti-inflammatory potency) between the micro-root extracts and the wild-grown plants were not clear, making it difficult to prove the medicinal equivalence of the in vitro culture products;
- Line 332: Lack of a clear definition of the “1 vvm aeration rate” for the temporary immersion system;
- The myricetin galactoside of the microshoot is labeled as “not detected” in Table 1, while the specific content is shown in Table 2, which needs to be further verified for data consistency;
- Listing information such as growth periods and geographic distribution of the wild-grown plants. Can data obtained from these plants in different years be reproduced? Is their growth periods same with those plants in the experimental groups?
- This study identifies multiple phenolic compounds through MS. Is it reliable to directly characterize substances solely based on m/z parameters? To combine spectral profiles and add reference standards, especially for abundant or notable phenolic substances would enhance the data reliability.
- The core variables are the culture type and the growth period in this study. Why not consider the culture medium composition? Regarding the growth period of 30 d, 60 d and 90 d, why not choose other time points? For example, in shaking culture, phenolic productivity already reached its highest point at the first time point of 30 d. What about conditions at 10 d, 20 d, or 40 d? It is suggested to increase the time gradient to fully verify that 30 d is indeed the time point with the highest phenolic productivity across all periods.
- In the shaking culture, phenolic productivity peaked at 30 days (19 mg/L/d) but declined sharply by 90 days (no specific value provided). Attributing this solely to "necrosis" without considering the metabolic regulatory is inappropriate, such as changes in phenolic synthase activity.
- Although the phenolic content in the temporary immersion system (26 mg/g DW) was lower than in shaking culture, it demonstrated greater stability. However, its potential advantages were not discussed, a comprehensive comparison of the advantages, disadvantages, and significance of each cultivation method at each time period is required.
- Are all phenolic substances mentioned in this study truly beneficial? Potentially harmful compounds that could complicate subsequent isolation of individual compounds might exist, as higher concentrations of all phenolics are not necessarily desirable. Furthermore, this study did not analyze whether the culture system might alter metabolic flux distribution—for instance, promoting catechin synthesis while suppressing other phenolic acids.
Minor points:
- Crucial details regarding key parameters are lacking, such as the specific model of the temporary immersion system used.
- The footnote for Table 1 needs to clearly add the detection limit (for instance, ND:<0.1 mg/g DW).
- Figure 3 lacks a scale bar in the culture status photos, making it impossible to assess micro-shoot size. Additionally, necrotic areas are not annotated with arrows.
- Figure 5 does not explain the rationale for selecting specific compounds to highlight, rather than displaying all detected metabolites.
- “microshoots grown in shake flasks for 90 days showed visible signs of necrosis in a submerged part of the biomass—a clear indicator of the culture entering a death phase”, the “death phase” should be revised to “decline phase”.
- This study lacks an in-depth exploration of the linkage mechanism between phenolic accumulation and aeration in the culture system (e.g., the induction of secondary metabolism by oxidative stress). Furthermore, it fails to explain contradictory findings—such as the temporary immersion system demonstrating superior results in some species, whereas it proved inferior to shaking culture in this specific study.
Author Response
Kindly find the attached document.

Reviewer 2 Report
Comments and Suggestions for Authors
Manuscript „Accumulation of Phenolic Compounds in Microshoot Cultures of Rhododendron tomentosum Harmaja (Ledum palustre L.)”
This study aimed to qualitatively and quantitatively evaluate phenolic substances in the Rhododendron tomentosum under different in vitro culture systems. For reference, the phenolic compositions of wild-grown R. tomentosum from two locations (Poland and Finland) were also analyzed. The results are interesting and worth promoting. However, the work needs to be corrected. Notably, the lack of molecular studies in IJMS is somewhat surprising.
My comments are as follows:
- The abstract should include information on the similarity of phenolic content to wild plants.
- The keywords contain repetition from the title (Ledum palustre).
- The “M&M” does not present enough details.
- What is the origin of tomentosum plants from which cultures were initiated?
- What is the control in the performed experiments?
- Please add what samples of wild plants were taken for analysis (stems, leaves?). When were samples taken?
- Results and Discussion
In each subsection, the results should be presented first, followed by a discussion.
Figures 2, 5, and 5 are unreadable and require correction. Please enlarge the font in axis titles and statistical indexes. If there are more letters in statistical indexes, please use a hyphen (e.g., b-e instead of bcde).
Figure 5 –please add information on the intact plants
Ref. 2.1- The description in lines 89-94 is outside the topic of the work. I suggest using it in the conclusions, if any.
Is it possible to use tissue cultures to produce bioactive substances of R. tomentosum, and what are the yields and costs?
Author Response
Kindly find the attached document.

Round 2
Reviewer 2 Report
Comments and Suggestions for Authors
It's just not clear what Figs 3 and A1 mean (lines 142 and 178).